# Multiple objects evoke fluctuating responses in several regions of the visual pathway

**Meredith N Schmehl**[1,2,3]*, **Valeria C Caruso**[4], **Yunran Chen**[5], **Na Young Jun**[1,3], **Shawn M Willett**[6], **Jeff T Mohl**[7], **Douglas A Ruff**[8], **Marlene Cohen**[8], **Akinori F Ebihara**[9], **Winrich A Freiwald**[9], **Surya T Tokdar**[3,5], **Jennifer M Groh**[1,2,3,10,11,12]

[1]Department of Neurobiology, Duke University, Durham, United States; [2]Center for Cognitive Neuroscience, Duke University, Durham, United States; [3]Duke Institute for Brain Sciences, Duke University, Durham, United States; [4]Department of Psychiatry, University of Michigan, Ann Arbor, United States; [5]Department of Statistical Science, Duke University, Durham, United States; [6]Department of Ophthalmology, University of Pittsburgh, Pittsburgh, United States; [7]American Medical Group Association, Alexandria, United States; [8]Department of Neurobiology, University of Chicago, Chicago, United States; [9]The Rockefeller University, New York, United States; [10]Department of Psychology & Neuroscience, Duke University, Durham, United States; [11]Department of Computer Science, Duke University, Durham, United States; [12]Department of Biomedical Engineering, Duke University, Durham, United States

*For correspondence: meredith.schmehl@duke.edu

**Abstract** How neural representations preserve information about multiple stimuli is mysterious. Because tuning of individual neurons is coarse (e.g., visual receptive field diameters can exceed perceptual resolution), the populations of neurons potentially responsive to each individual stimulus can overlap, raising the question of how information about each item might be segregated and preserved in the population. We recently reported evidence for a potential solution to this problem: when two stimuli were present, some neurons in the macaque visual cortical areas V1 and V4 exhibited fluctuating firing patterns, as if they responded to only one individual stimulus at a time (Jun et al., 2022). However, whether such an information encoding strategy is ubiquitous in the visual pathway and thus could constitute a general phenomenon remains unknown. Here, we provide new evidence that such fluctuating activity is also evoked by multiple stimuli in visual areas responsible for processing visual motion (middle temporal visual area, MT), and faces (middle fundus and anterolateral face patches in inferotemporal cortex – areas MF and AL), thus extending the scope of circumstances in which fluctuating activity is observed. Furthermore, consistent with our previous results in the early visual area V1, MT exhibits fluctuations between the representations of two stimuli when these form distinguishable objects but not when they fuse into one perceived object, suggesting that fluctuating activity patterns may underlie visual object formation. Taken together, these findings point toward an updated model of how the brain preserves sensory information about multiple stimuli for subsequent processing and behavioral action.

## Editor's evaluation

This important study adds to the growing body of evidence that neural responses fluctuate in time to alternatively represent one among multiple concurrent stimuli and that these fluctuations seize when objects fuse into one perceived object. The present study provides solid evidence from

multiple brain areas and stimuli types to support this hypothesis. Overall, the study illustrates how the brain can use time dimension and synchrony to either parse or integrate stimuli into a coherent representation.

## Introduction

Neurons are broadly tuned. For example, even in primary visual cortex, neurons typically have receptive fields that are several times larger than spatial acuity at the perceptual level (*Dow et al., 1981*; *Smith et al., 2001*; *Motter, 2009*; *Freeman and Simoncelli, 2011*; *Grill-Spector et al., 2017*; *Keliris et al., 2019*). Due to this broad tuning, a given neuron may be capable of responding to more than one sensory object in a scene. When multiple stimuli are present, how does the brain preserve information about each item?

We recently reported a potential solution to this problem: When two stimuli co-occur, individual neurons may fluctuate between responding to only one or the other stimulus at a time, allowing dynamic populations of neurons to encode different stimuli at the same time. In our earlier work, we developed statistical approaches to evaluate whether the fluctuating activity patterns predicted from such a scheme exist (*Caruso et al., 2018*; *Mohl et al., 2020*; *Glynn et al., 2021*), and we reported evidence for this phenomenon in both auditory (*Caruso et al., 2018*) and visual (*Caruso et al., 2018*; *Jun et al., 2022*) areas. In particular, we found evidence for fluctuating activity in the inferior colliculus (*Caruso et al., 2018*), primary visual cortex (V1), area V4 (*Jun et al., 2022*), and the middle fundus (MF) face patch of inferotemporal (IT) cortex (*Caruso et al., 2018*) when two distinguishable stimuli were presented. Relatedly, oscillatory patterns of activity triggered by the appearance of multiple stimuli have also been observed in V4 using other paradigms and analysis methods (*Kienitz et al., 2018*; *Kienitz et al., 2022*).

In short, activity fluctuation has now been identified in a wide array of brain areas, from early auditory regions to early, middle, and late visual cortical regions. This breadth of brain areas suggests that fluctuating activity may reflect a general mechanism of neural representation supporting the encoding of multiple stimuli. However, the relevant factors that govern the occurrence of such fluctuating activity remain poorly understood. Here, we explore how fluctuating activity may depend on the nature of the stimuli involved. In particular, we investigate the roles of (1) neural selectivity for stimulus type and (2) the discriminability of stimuli into one vs. more than one object. In so doing, we expand the scope of the visual brain regions studied to include the middle temporal area (MT) and multiple areas of the IT face patch system (the MF, and anterolateral, AL, face patches).

We report that neuronal activity in all three areas fluctuated across trials between responses that appeared to be drawn from the response distributions corresponding to the respective individual stimuli. In MF and AL, fluctuating neuronal activity was present when two faces were presented (i.e., stimuli for which the patches are specialized) as well as when a face and a non-face object were presented, but in MF the fluctuations were more common when two faces were presented (i.e., two stimuli of the category for which the neurons appear specialized) than when a face and a non-face object (different stimulus categories) were presented. This suggests that multiple faces compete more severely for representation in this face-specific system than do non-face objects. In MT, such fluctuating activity was present only when the two visual stimuli were distinct (adjacent gratings or dot patches), and not when they were superimposed and fused to form single objects (plaids), thus implicating fluctuating activity as playing a role in perceptual segregation of distinct visual elements into distinct objects, supporting previous findings in V1 (*Jun et al., 2022*).

Together, these findings suggest that fluctuating activity patterns are a widespread phenomenon connected to the computations underlying object formation and identification. Object separation appears generally necessary for fluctuating activity to occur, and specialization for certain types of stimuli can also affect the prevalence of fluctuations across the neural population.

## Materials and methods
### Introduction to datasets

We analyzed multiple datasets from two visual cortical regions (MT and IT cortex). All datasets were collected in awake, behaving macaques (MT: *Macaca mulatta*; IT: *Macaca mulatta* and *Macaca*

**Table 1.** Comparison of datasets.

| Dataset name | Citation | Brain area | Stimuli | Stimulus locations | Task | Recording method |
|---|---|---|---|---|---|---|
| MT adjacent gratings | *Ruff and Cohen, 2016b* | MT | Drifting images of two gratings | Adjacent (3 degrees eccentric) | Detect orientation change of a different, ipsilateral stimulus | Array, single and multi-unit |
| MT dot patches | *Li et al., 2016* | MT | Moving dots | Adjacent | Detect change in fixation stimulus | Single electrode, single unit |
| MT plaid | *Ruff et al., 2016a* | MT | Drifting images of gratings and plaids | Same location | Passive fixation | Array, single and multi-unit |
| IT face patch MF | *Ebihara, 2015*; *Caruso et al., 2018* | MF | Images of faces and objects | Adjacent | Passive fixation | Single electrode, single unit |
| IT face patch AL | *Ebihara, 2015* | AL | Images of faces and objects | Adjacent | Passive fixation | Single electrode, single unit |

*fascicularis*). The data were originally collected at the University of Pittsburgh (*Ruff and Cohen, 2016b*; *Ruff et al., 2016a*), Rockefeller University (*Ebihara, 2015*), and the German Primate Center in Goettingen (*Li et al., 2016*). All procedures were conducted in accordance with applicable US, German, and European guidelines or regulations and were approved by the relevant compliance committees at the relevant institutions (US guidelines: National Institutes of Health (NIH) Pub. No. 86-23, Revised 1985; Pittsburgh MT grating datasets: Institutional Animal Care and Use Committees of the University of Pittsburgh and Carnegie Mellon University (Protocol #: 20067560, PHS Assurance Number: D16-00118); IT datasets: The Rockefeller University Institutional Animal Care and Use Committee (Protocol #: 21,104 H USDA, PHS Assurance Number: A3081-01); Goettingen MT dot patch dataset: [Niedersächsisches Landesamtfür Verbraucherschutzund Lebensmittelsicherheit (LAVES)] permit numbers 33.42502/08-07.02 and 33.14.42502-04-064/07). All surgical procedures were conducted aseptically and with suitable anesthetics and analgesics.

Our goal was to assess fluctuating neural activity in multiple visual areas and in response to multiple stimuli (*Table 1*). In all datasets, fixation was controlled while the stimuli were presented (0.5–1.2 degree fixation window radius, depending on the dataset). In some datasets, monkeys performed a behavioral task in which attention was directed away from the stimuli, such as to the fixation position itself or to an additional stimulus in the ipsilateral hemifield. In the fixation-only tasks, it should be noted that the absence of experimental control over attention could have led to the monkeys covertly attending the receptive field stimuli from time to time.

For area MT, we used three datasets. The first dataset included single- and multi-unit responses to drifting grating stimuli positioned adjacent to one another (*Figure 1a.1 and 2*; *Ruff and Cohen, 2016b*). The experiment involved simultaneous recordings from V1, V4, and MT. As we have previously identified fluctuating response patterns in V1 and V4 (*Jun et al., 2022*), this MT dataset provides an opportunity to compare the presence of fluctuating activity across visual areas under the same experimental conditions (and some of the same recording sessions) as in our previous work. However, this dataset was comparatively small (as detailed further below), making additional data desirable.

Accordingly, we included a second dataset from a previously published study involving single- and multi-unit responses to adjacent random dot stimuli (*Figure 1b*; *Li et al., 2016*). Both of these datasets were recorded while monkeys performed a behavioral task in which attention was directed away from the stimuli in the MT receptive field: toward the opposite hemifield (*Ruff and Cohen, 2016b*) or to the fixation point (*Li et al., 2016*). Fixation was maintained during stimulus presentation to within 0.5 degrees of the fixation point for the Ruff and Cohen dataset and 1.2 degrees for the Li et al. dataset.

In both of these MT datasets, the stimuli can be considered as distinct and distinguishable objects from each other, regardless of whether they are gratings or patches of moving dots, because they are presented at separate locations. Our third MT dataset provides a comparison to these first two. In this dataset, individual gratings were presented at the same location in space, such that when they were presented together, they fused to form a plaid stimulus (*Figure 1a.1 and 3*). This dataset was originally published in *Ruff et al., 2016a* and involved simultaneous V1 and V4 recordings previously analyzed for fluctuating activity in *Jun et al., 2022*. In this dataset, monkeys performed a simple fixation task.

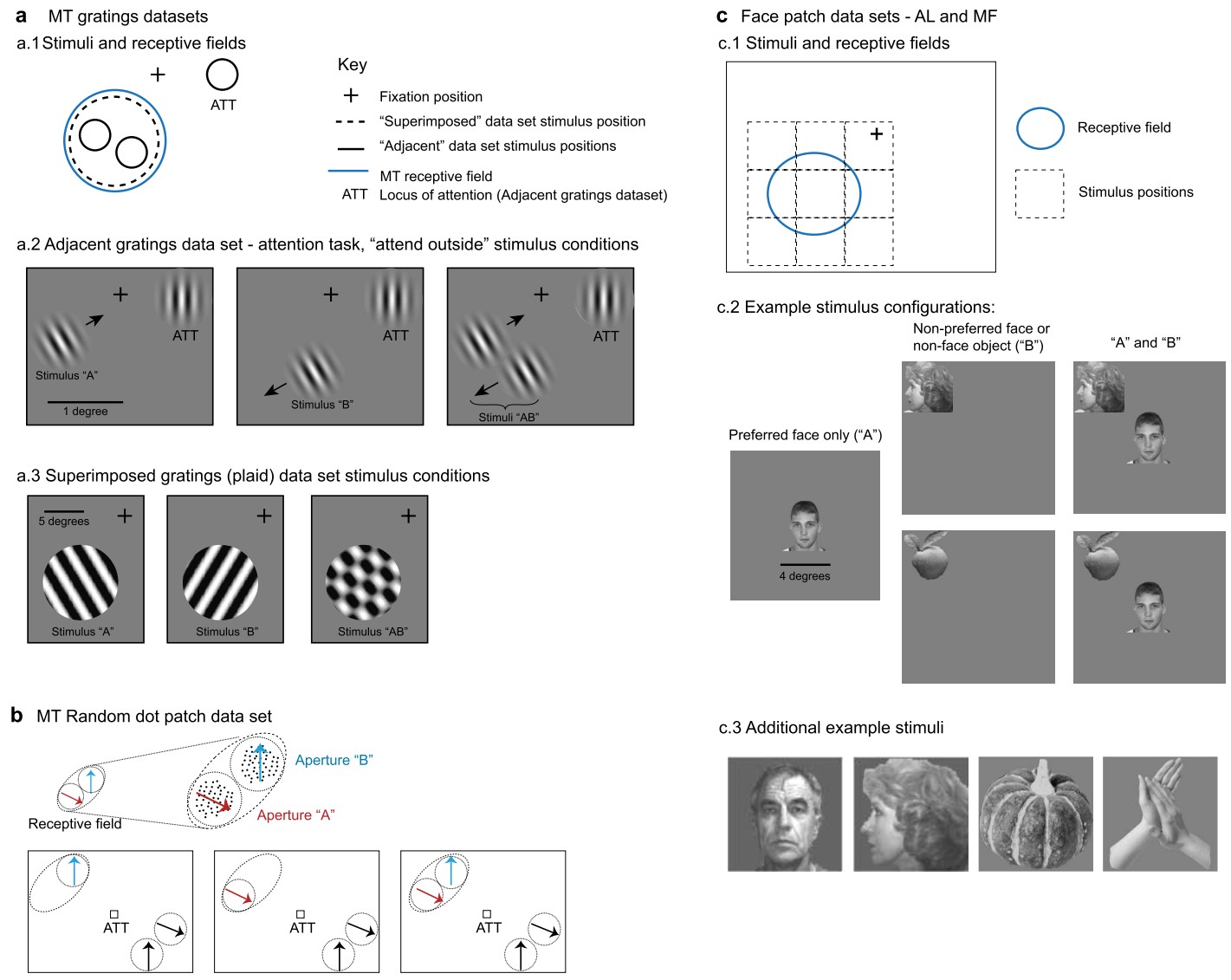

**Figure 1.** Description of datasets. (**a**) Middle temporal area (MT) gratings datasets from *Ruff and Cohen, 2016b* and *Ruff et al., 2016a*. (**a.1**). For the adjacent gratings, the two stimuli were placed side by side and roughly within the MT neuron's receptive field. The superimposed gratings were larger and each was placed at the same position, also within the MT neuron's receptive field. (**a.2**) Monkeys performed an attention task during the presentation of the adjacent gratings; only trials in which attention was successfully directed to the stimulus in the ipsilateral hemifield were included in this study. (**a.3**) Monkeys performed a fixation task during presentation of the superimposed gratings. (**b**) MT random dot patch dataset from *Li et al., 2016*. Patches of moving dots were presented within apertures placed within the receptive field of an MT neuron. Attention was directed to the fixation target for the trials included in the present study. (**c**) Face patch stimulus conditions (*Ebihara, 2015*). (**c.1**) Monkeys performed a fixation task and stimuli were presented either at the center of the receptive field or from one of eight locations surrounding it. (**c.2**) 'Preferred' faces were faces that evoked a strong response, and were presented in the center of the receptive field. Non-preferred faces or non-face objects were presented at one of the other locations. Combinations of stimuli involved the preferred face in the center and one of the non-preferred stimuli at one of the adjacent locations. (**c.3**) Additional examples of stimuli used in the face patch datasets.

The online version of this article includes the following figure supplement(s) for figure 1:

**Figure supplement 1.** Properties of the datasets and application of exclusion criteria.

---

The specific stimulus details for these MT datasets are as follows. In the adjacent gratings dataset, the grating sizes were selected to lie within the MT receptive field under study and did not exceed 1 degree in diameter. The gratings used in the superimposed dataset were larger, ranging between 7 and 13 degrees in diameter, and covered the MT receptive field. The sizes of the random dot patches

are not available, but they were depicted at adjacent positions within the MT receptive field (*Li et al., 2016*).

For the IT cortex, we included datasets involving single-unit recordings in two areas of the face patch system, areas MF and AL, during a fixation task (*Figure 1c*; fixation maintained within 1 degree) (*Ebihara, 2015*). We have previously reported fluctuating activity in face patch MF (*Caruso et al., 2018*), but here we have re-analyzed the MF dataset (see 'Analysis: response periods and inclusion criteria') to provide detailed results for different types of stimuli (face–face vs. face–object pairs), allowing a comparison of results in the two face patches (MF and AL) in the same experimental conditions.

For both MF and AL, we assessed neural responses to image pairs drawn from three visual stimuli selected from a pool of faces and objects: a 'preferred face' (strong neural response), a 'non-preferred face' (little to no neural response), and a 'non-preferred object' (little to no neural response). These stimuli were randomly presented either alone or in pairs, with each pair including the preferred face, positioned at the center of the neuron's receptive field, and one of the non-preferred stimuli, presented adjacent to the preferred face at one of eight equidistant locations. Stimuli were 4 × 4 degrees in size.

As in *Caruso et al., 2018*, we did not incorporate the exact location of the non-preferred face/object into the present analysis, but any excessive heterogeneity in the responses of the non-preferred stimuli due to location would likely result in exclusion of the condition from further analysis (see 'Analysis: response periods and inclusion criteria').

## Analysis: response periods and inclusion criteria

### Response periods

For each dataset, we compared neural responses (see 'Data analysis') using a spike counting window of 200 ms, the minimum stimulus presentation window across all datasets. This window was shifted by the approximate response latency of the brain region, specifically 30 ms for MT and 50 ms for IT, that is, spikes were counted in a window from 30 to 230 ms after stimulus onset for MT or 50–250 ms for IT. For the IT datasets, the stimuli were presented for 400 ms, and results were similar when we analyzed the data in a 50–450 ms time window.

### Trial inclusion criteria

Only correctly performed trials were analyzed. For fixation tasks, this meant that the monkey maintained fixation for the required interval (MT plaid dataset, IT datasets). For attention tasks, this meant that the monkey correctly identified the change in the attended stimulus, which was outside the receptive field of the neurons under study (MT gratings dataset: the motion change in a grating presented in the ipsilateral field; MT dot patch dataset: the change in color in the fixation target). In addition, trials were excluded from the MT gratings dataset if any microsaccades were identified during the stimulus presentation period. Microsaccades were defined as periods of time in which eye speed exceeded the mean eye speed in the fixation epoch by more than 6 standard deviations, following standards established by *Engbert and Kliegl, 2003* and deployed in our previous studies involving these same experiments (*Ruff et al., 2016a*; *Ruff and Cohen, 2016b*; *Jun et al., 2022*).

## 'Triplet' definition and inclusion criteria

Our analysis of fluctuating activity patterns relies on comparing the activity evoked on trials in which two stimuli, A and B, are presented simultaneously (AB) to trials in which only the A or only the B stimulus was presented. We refer to the corresponding A-alone, B-alone, and combined AB conditions as 'triplets'.

To be included for analysis, triplets were required to meet several criteria. First, the minimum number of included trials for each of the A, B, and AB components was set at five trials for each, in keeping with our prior work and assessments of the sensitivity of the analysis to the number of trials involved (*Caruso et al., 2018*; *Mohl et al., 2020*; *Jun et al., 2022*). The average numbers of trials of the included triplets across datasets are illustrated in *Figure 1—figure supplement 1*. The two MT datasets involving adjacent stimuli had the lowest average trial counts (average of the A, B, and AB trial counts was ~18 trials for the gratings and ~6 trials for the dot patches), whereas the MT 'plaid' dataset had the most (average ~37 trials; all values computed after imposition of the full set of exclusion criteria, including those described below).

Second, spike count distributions for the A- and B-alone trials in a triplet were required to be suffi-
ciently well described by a Poisson distribution (with a variance roughly equal to the mean). The spike
count distributions for A and B trials for triplets in all five datasets exhibited variance-to-mean ratios
(Fano factors) with a mode of about 1, but there was some spread around that value (*Figure 1—figure
supplement 1B*). Accordingly, we discarded triplets for which the average variance-to-mean ratio in
the A and B conditions was greater than 3. This value was chosen to exclude the worst outliers while
preserving adequate data across all datasets to allow for appropriate comparisons. Screening with a
more stringent criterion (variance-to-mean ratio <2) reduced the overall quantity of data but did not
change the key results we report here (*Figure 4—figure supplement 1*). Notably, screening with the
variance-to-mean ratio is a deviation from our previous approach involving an approximate chi-square
goodness of fit test with Monte Carlo p-value calculation (*Caruso et al., 2018*; *Jun et al., 2022*).
The former approach had the undesired effect of being more likely to exclude conditions with larger
amounts of data, whereas the present method is neutral to dataset size.

Finally, spike count distributions for the individual A and B stimuli in a triplet were required to
be sufficiently separable from one another. This is necessary so that it is possible to ascertain with
some confidence how the AB spike count distributions compare to each of the corresponding single-
stimulus A and B distributions. We set the inclusion criterion as a 95% likelihood that the A and B
spike counts were drawn from different Poisson distributions (specifically, the logarithm of the intrinsic
Bayes factor was required to be greater than 3, given an a priori 50–50 chance that the two distribu-
tion means are the same vs. different). The greater numbers of trials in some datasets than in others
allowed for inclusion of triplets at smaller differences in the A vs. B response distributions (*Figure 1—
figure supplement 1*), but this did not appear to impact the overall conclusions (see *Figure 1—figure
supplement 1* and associated legend).

## Data analysis

A full description of our statistical method has been published previously (*Caruso et al., 2018*; *Mohl
et al., 2020*; *Jun et al., 2022*). Our method centers on modeling spike counts based on Poisson distri-
butions, a common technique for handling non-negative count data in neuroscience and other fields.

Briefly, we compare the distribution of spike counts in response to a stimulus pair (AB) to the distri-
butions of spike counts in response to the component stimuli presented individually (A and B). We
consider four possibilities for the relationship between the AB response and the individual A and B
responses (*Figure 2*). The first three attempt to fit the AB responses with a single Poisson distribution,
as follows:

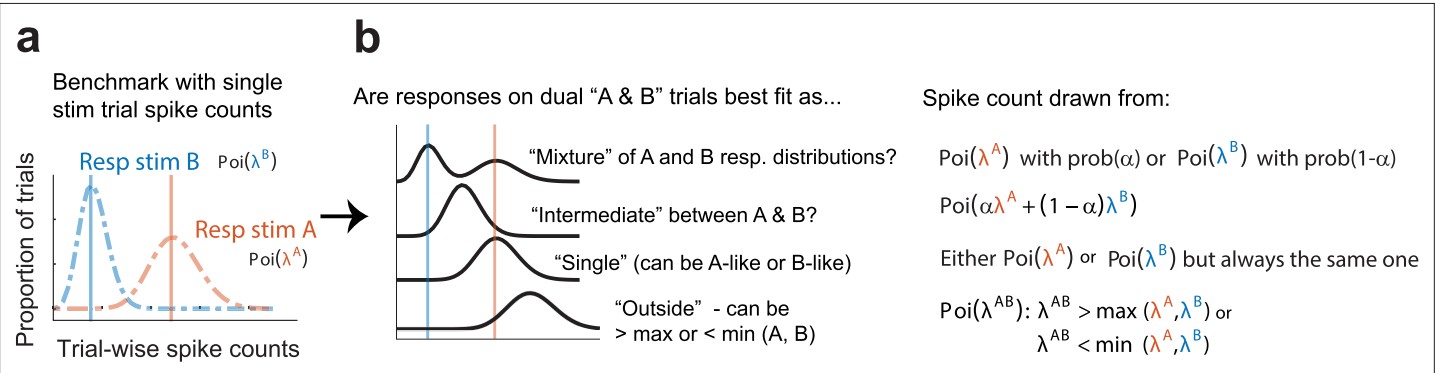

**Figure 2.** Statistical comparisons to be made across datasets. We assign each trial into a 'triplet' consisting of trials in which stimulus A (red), stimulus
B (blue), or both stimulus A and B (AB, black) were presented. (**a**) Within each triplet, spike count distributions are compared to determine how the
AB distribution relates to the A and B distributions. (**b**) The AB spike count distribution may be: a mixture of the A and B distributions, suggesting
fluctuations between the responses to the two stimuli; an intermediate rate between the A and B distributions, suggesting averaging (or fluctuations
on a faster time scale than the analysis period); a rate that matches either the single A or B distribution, suggesting a winner-take-all strategy; or a rate
outside the range bounded by the A and B distributions, suggesting addition or subtraction of the responses. Spike counts are assumed to be drawn
from Poisson distributions, with a weighting parameter alpha defining the relative contribution of the individual A and B distributions in 'mixtures' and
'intermediates'.

### Outside
The mean of the distribution of AB responses might be higher than the maximum or lower than the minimum of A or B. This possibility includes either summation of the mean A and B responses or strong suppression of responses to one stimulus by the presence of the other.

### Single
The mean of the distribution of AB responses might match the mean response to either A or B. This is akin to winner-take-all, with the same winner on every trial.

### Intermediate
The AB responses might be well fit by a single Poisson distribution with a mean between A and B, as if the neuron performed an averaging operation on its inputs.

These possibilities are in contrast to the final possibility of interest:

### Mixture
The AB responses match A on some trials and match B on other trials, creating a mixture of the two A and B distributions across trials (fluctuations or 'multiplexing' from trial to trial). We model this possibility as a bimodal mixture of the A and B Poisson distributions as assessed on those single-stimulus trials.

We evaluated these four hypotheses by computing their posterior probabilities based on intrinsic Bayes factors (*Berger and Pericchi, 1996*; *Caruso et al., 2018*). We used equal priors (25%) for each of the four possibilities. For the purposes of this work, we defined 'fluctuating responses' as responses that were classified as 'mixture' with a posterior probability greater than 0.67 (i.e., mixture is at least twice as likely as the other three categories combined).

As noted previously (*Caruso et al., 2018*; *Jun et al., 2022*), these category boundaries may not always perfectly capture the presence of fluctuating activity. For example, if a neuron sometimes switches between responding to A and B stimuli in the middle of a stimulus presentation, it will tend to be classified in the 'intermediate' bucket. Similarly, if a neuron responds to A on 90% of the trials, it is likely to be classified as 'single' even if it responds to B on the remaining 10%.

## Results

### Spiking activity fluctuates in MT when two stimuli are distinct, but not when they fuse into one object

We first evaluated the responses in visual area MT to stimuli that appeared as distinct objects. The first dataset involved adjacent drifting gratings, both of which were positioned within the receptive field of the MT neuron under study. These data were recorded simultaneously with recordings in V1, in which we have previously reported the presence of fluctuating activity in response to these stimuli (*Jun et al., 2022*). On the trials included for analysis, monkeys performed a task in which attention was directed elsewhere.

As in V1, we found that some units exhibited bimodal AB spike count distributions whose modes approximately match the individual A and B means, suggesting that the AB responses fluctuate between the A and B response rates across trials. An example of such a unit is shown in *Figure 3a*: the red and blue dashed lines indicate the means of the A- and B-alone response distributions, and the black curve provides a smoothed depiction of the distribution of spike counts across the combined AB trials. This distribution appears to reflect an admixture of the A- and B-alone response distributions, indicating that the unit fluctuates between each single response rate from trial to trial.

We next considered a second independent dataset from MT (*Li et al., 2016*) that shares similar properties to the adjacent gratings dataset. In this study, two separate patches of dots, each moving in different directions, were presented within the receptive field of the neurons under study. For the included trials, the monkey was performing a task in which attention was directed to a fixation point away from the dot patches. The combined stimulus AB trials in this dataset were previously reported to show fluctuating activity patterns using a different analysis method (*Li et al., 2016*), and the dataset was made publicly available at *Li et al., 2016*. As in the adjacent gratings MT dataset, we observed

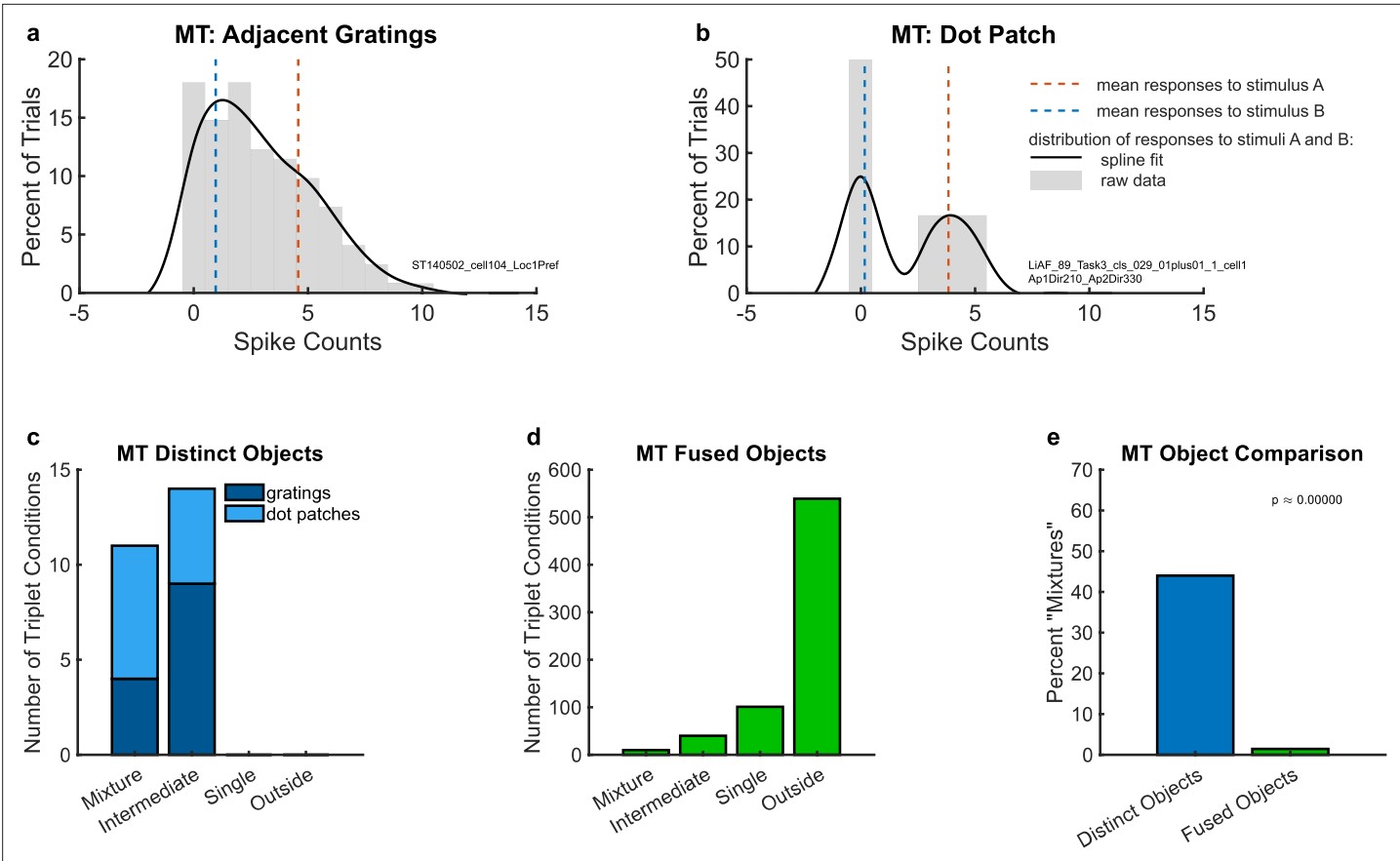

**Figure 3.** Example individual unit responses and population results for middle temporal area (MT). (**a**) An example unit's response to two simultaneous adjacent gratings (AB, black solid lines) in comparison to the mean response to each grating individually (A and B, red and blue dotted lines). (**b**) A different example unit's response to simultaneous dot patches. In both (**a**) and (**b**), gray bars indicate the raw distributions. The black lines indicate a smoothed fit for display purposes; this fit was generated by convolution with a 3-point sliding window of which the middle point is weighted twice as much as either of the two outer points followed by cubic spline interpolation. Spike counts were computed for a window 200 ms in duration (see Materials and Methods); multiply x-axis by 5 to convert to spike rates in Hz. (**c**) Number of triplets classified into each category (as shown in *Figure 2*) when responses to two simultaneous gratings (dark blue) or dot patches (light blue) are recorded in MT. Only triplets for which the winning model garnered a posterior probability of 0.67 or greater are shown. (**d**) Same as (**c**) but for fused objects, created by overlaying two gratings to form a plaid. (**e**) Comparison of (**c**) and (**d**) showing that fluctuating (mixture) responses are only present when two objects are distinct, and not when two objects are fused into one. Results in this figure only include triplet conditions for which the average variance-to-mean ratio (Fano factor) of activity on the A- and B-only trials was less than 3. See *Figure 4—figure supplement 1* for corresponding findings using stricter inclusion criteria.

bimodal spike count distributions on combined AB trials with modes that appeared to match the mean responses observed on A- and B-alone trials (*Figure 3b*).

To evaluate the prevalence of such fluctuating activity patterns at the population level, we deployed the statistical comparison described in Materials and methods, in which spike count distributions on combined AB trials were evaluated in comparison to four possible options: 'mixture', 'intermediate', 'single', and 'outside' (*Figure 2*). *Figure 3c* shows the results of this analysis for the adjacent gratings and dot patches datasets. In these two datasets, we only identified 'mixtures' and 'intermediates', but no 'singles' or 'outsides'. There was little difference between these two datasets (light blue vs. dark blue stacked bars in *Figure 3c*). Overall in these two datasets, 'mixtures' constituted about 45% of the categorized conditions (*Figure 3e*, bar labeled 'distinct objects').

In contrast, evidence of fluctuating activity was much lower in our third MT dataset, which involved gratings that were either presented individually (A- and B-alone) or superimposed to produce a fused percept of a 'plaid' or checkerboard-like pattern. As in the first dataset, the stimuli were presented in the receptive field of the MT neuron under study, but this time they were presented at the same exact location, rather than adjacent to one another. The monkeys were performing a fixation task; no explicit attentional report was required. In this dataset, 'mixture' responses were very rare, constituting only

about 1.4% of the conditions that were well categorized (*Figure 3d, e*). The majority of conditions were categorized as 'outside' (~78%), with 'single' and 'intermediate' accounting for a further ~14% and ~6%, respectively.

The difference in the proportions of 'mixture' response patterns in the first two MT datasets involving distinct objects (~45%) vs. this last dataset in which the two stimuli fuse to form a new third object (~1.4%) was statistically significant (*Figure 3e*, blue vs. green bars, chi-squared test p < 10⁻⁹). This finding is consistent with the interpretation that the prevalence of fluctuating activity is related to whether two stimuli are perceptually distinguishable vs. fuse to form a single object.

## Spiking activity fluctuates in face patches MF and AL when two stimuli are distinct

To further explore the generality of these fluctuating activity patterns, we next turned to a higher order area of the visual pathway – the face patch system of IT cortex. Face patches are exquisitely sensitive to the identity of individual faces, and respond preferentially to faces as compared to other objects. This affords the opportunity to test whether fluctuating activity is evoked solely when the two stimuli are both faces, as might be expected because the non-face objects are largely thought to be encoded by neural populations in other regions of IT, or if both face–face and face–object pairs evoke fluctuating activity. We previously identified fluctuating activity patterns in the MF face patch (*Caruso et al., 2018*), but we did not separate face–face and face–object stimulus conditions. Here, we revisited that dataset to conduct that comparison. We also sought to compare the results in MF to those in face patch AL. Recent work has suggested that MF is 'earlier' in the face patch hierarchy than AL, and that AL shows properties such as view-invariance that are less evident in MF (*Moeller et al., 2008*; *Freiwald and Tsao, 2010*; *Ebihara, 2015*). Thus, any differences between the results in MF and AL could shed light on how fluctuating activity varies within the processing hierarchy at this late stage of visual processing.

Accordingly, we assessed MF and AL face patch responses with both face–face and face–object stimulus combinations. In face–face combinations, one face evoked the strongest response from among a stimulus set (preferred face), and the other face evoked the weakest response from the same set. The preferred face was presented at the center of the neuron's receptive field, while the non-preferred face was presented at a different location (*Figure 1c*). Similarly, the object stimuli were also chosen to evoke weak responses and were presented at a different location than the preferred face. The stimulus conditions were randomly interleaved, and the data were collected while monkeys performed simple fixation tasks.

Both face patch regions and both types of stimulus combinations elicited fluctuating activity. *Figure 4a–d* shows the responses of four example neurons, each of which exhibited spike counts that fluctuated across trials between appearing to respond to the preferred vs. non-preferred face (*Figure 4a, c*) or to the face vs. the object (*Figure 4b, d*).

To evaluate these results at the population level, we used the same statistical comparison described above for the MT analysis. As shown in *Figure 4e*, we identified 'mixtures' in face patch MF for both face–face (dark brown bar) and face–object (mustard-colored bar) stimulus combinations. We also observed 'intermediates' and 'singles', but no 'outsides'. Overall, 'mixtures' constituted about 32.5% of well-categorized conditions (*Figure 4g*, light brown bar). The proportion of 'mixtures' was higher for face–face stimulus combinations (43.6%, *Figure 4g*, dark brown bar) than for face–object stimulus combinations (22.7%, *Figure 4g*, mustard-colored bar). The difference in these proportions was just barely significant by chi-squared test (p = 0.043), but we note that when using stricter Fano factor inclusion criteria for the dataset (variance-to-mean ratio of single-stimulus responses less than 2 rather than 3, see Materials and methods: 'Triplet' definition and inclusion criteria), the reduction in the amount of included data increases this p value to p = 0.054 (see *Figure 4—figure supplement 1*). Accordingly, this aspect of the results should be viewed as of borderline significance.

'Mixtures' were even more prevalent in face patch AL (*Figure 4f*), constituting about 66% of the conditions for both face–face and face–object pairs (*Figure 4g*). The approximate doubling in the prevalence of 'mixtures' in AL vs. MF (65.9 vs. 32.5%) was statistically significant by chi-squared test (p = 0.00002). Like face patch MF, AL responses had no 'outsides', but 'singles' and 'intermediates' were still present. In MF, we previously showed that such 'intermediates' could show fluctuating activity within the timescale of individual trials (*Caruso et al., 2018*).

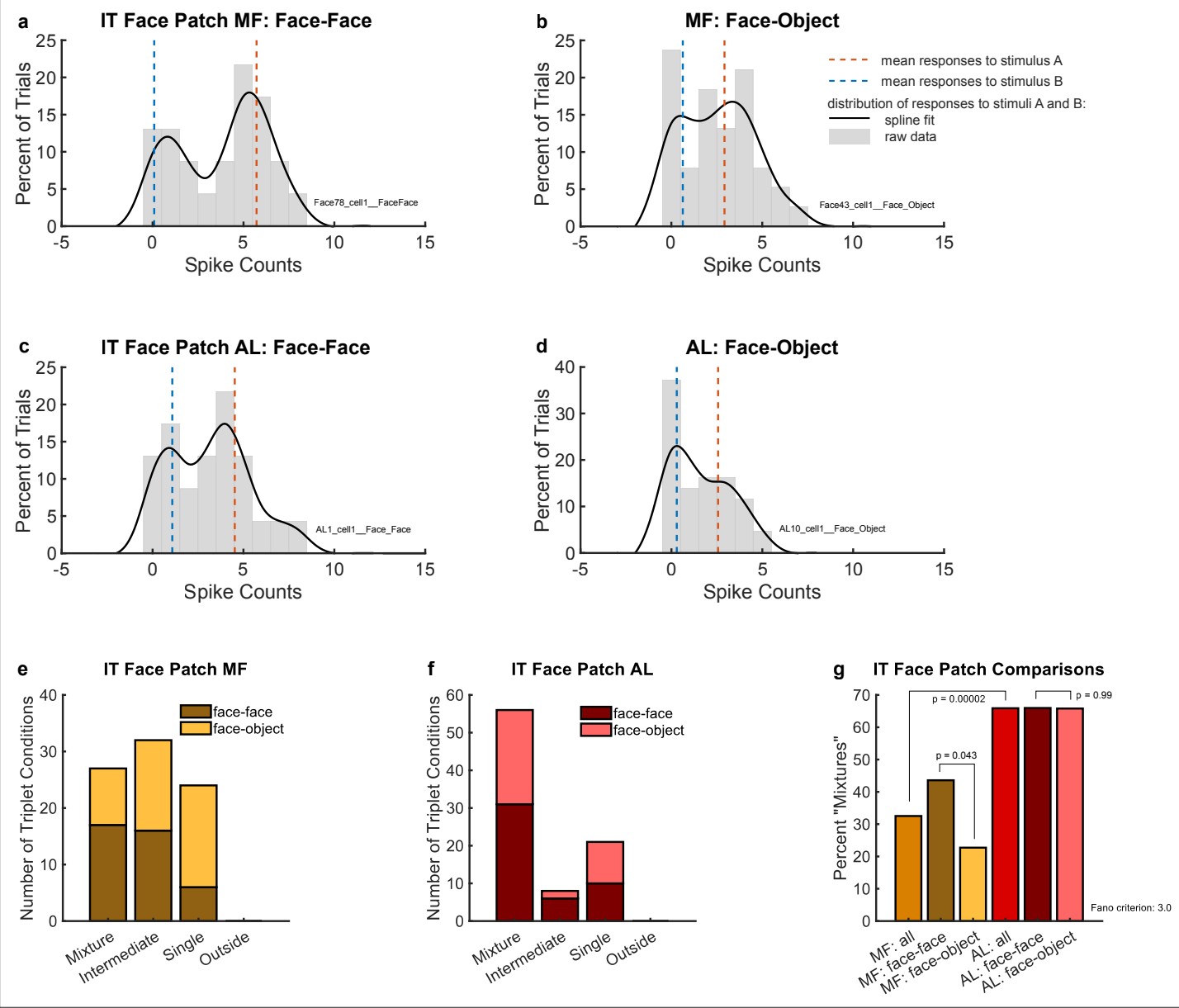

**Figure 4.** Example individual unit responses and population results for IT face patches middle fundus (MF) and AL. (**a**) An example single MF unit's response to two simultaneous faces (AB, black solid lines) in comparison to the mean response to each face individually (A and B, red and blue dotted lines). (**b**) Same as (**a**) but in response to one face and one object. (**c**) Same as (**a**) but recorded in AL. (**d**) Same as (**b**) but recorded in AL. In (**a**) through (**d**), distributions were smoothed for display purposes via convolution with a 3-point sliding window of which the middle point is weighted twice as much as either of the two outer points, followed by cubic spline interpolation. Gray bars show the raw spike counts for the 200 ms spike counting windows (multiply *x*-axis scale by 5 to convert to spike rates in Hz). (**e**) Number of triplets classified into each category (as shown in *Figure 2*) when responses to a face–face pair (dark brown) or a face–object pair (mustard color) are recorded in face patch MF. Only triplets for which the winning model garnered a posterior probability of 0.67 or greater are shown. (**f**) Same as (**e**) but recorded in face patch AL. (**g**) Comparison of (**e**) and (**f**) showing that more conditions display a fluctuating (mixture) pattern in AL than in MF. In MF, there were more 'mixtures' among face–face pairs than among face–object pairs. Results in this figure only include triplet conditions for which the average variance-to-mean ratio (Fano factor) of activity on the A- and B-only trials was less than 3. See *Figure 4—figure supplement 1* for corresponding findings using stricter inclusion criteria.

The online version of this article includes the following figure supplement(s) for figure 4:

**Figure supplement 1.** Same as population results in *Figures 3 and 4*, but using a Fano factor criterion of 2 instead of 3.

## Discussion

A key problem in neuroscience is how the brain retains information about multiple co-occurring stimuli. We present a potential solution to this problem: When multiple stimuli are present, neurons across levels of visual processing may fluctuate, on the whole trial time scale, between responding to individual stimuli in distinct time windows. This fluctuating pattern may be a way for the brain to preserve information about multiple stimuli in complex sensory environments.

We report here that fluctuating activity was evoked by pairs of distinguishable stimuli in MT and areas MF and AL in the face patch system of IT cortex. This work extends our previous findings in several key ways. First, it expands the range of brain regions and stimulus conditions in which this phenomenon has been observed. In MT, fluctuating activity was observed in two independent datasets involving different stimulus conditions and collected by different labs. In the face patch system, fluctuating activity was observed in both face patches included in this study. This suggests that fluctuating activity evoked by combinations of stimuli is a robust phenomenon within the visual pathway and occurs across multiple stages of the visual hierarchy.

Second, another of our goals was to evaluate whether fluctuating activity is more prevalent when the two stimuli involved are competing for representation in the same neural population. The use of face–face and face–object stimulus combinations to probe signals in the face patch system provides an opportunity to address this question in a unique way. If the face patch system serves as a series of structures whose specific role is to represent faces and not non-face objects, one might expect that fluctuating activity would be more evident when the two stimuli were both faces than when only one of the stimuli is a face. Specifically, one might expect winner-take-all for the face stimulus, which would be classified as 'single' in our analysis. Instead, we observed fluctuating activity for both face–face and face–object pairs, although there was a modestly higher incidence of fluctuating activity for face–face stimuli than face–object stimuli in MF. In AL, fluctuating activity was very common for both types of stimulus combinations. This suggests that fluctuating activity can occur even when somewhat distinct populations of neurons are available to respond to the two stimuli in question; neurons outside the face patch system are in principle available to encode the presence and properties of the non-face object, yet fluctuating activity nevertheless occurred.

The presence of fluctuations evoked by face–object stimuli is reminiscent of a related aspect of our previous findings in V1 (*Jun et al., 2022*). Specifically, we observed fluctuating activity in V1 even though the two stimuli involved were usually not located within the same receptive field and in principle could have evoked activity in largely distinct neural subpopulations. One possible explanation for this is that the activity of the whole population is necessary to specify the presence or identity of a stimulus, even if some members of the population are silent. A second possibility is that fluctuations occur 'on spec' regardless of the particular stimuli involved due to the possibility that they may require activity in overlapping populations of neurons to be encoded. Supporting these possibilities, we found fluctuating activity not only in the two MT datasets, which involved adjacent stimuli in the same neural receptive field, but also in the IT and earlier V1 datasets, which largely did not.

Finally, the evidence for fluctuating activity in MT when the two stimuli were perceptually distinct but not when they fused to form a common object lends support to the idea that there is an interplay between segregation of the scene into separate objects and the occurrence of fluctuating activity patterns. We previously found a similar pattern in V1 although not in V4 (*Jun et al., 2022*). It is not yet clear what might account for the discrepancy with V4, and this matter bears further investigation.

There are both strengths and limitations to the use of multiple datasets across different laboratories, brain areas, and experimental conditions. The key strength is that the presence of fluctuating activity under a wide variety of circumstances indicates that this property of neural representations is a robust one that does not require specialized circumstances to uncover. Across this and our two previous studies (*Caruso et al., 2018*; *Jun et al., 2022*), we have now shown fluctuating activity occurring in the context of behavioral tasks requiring a report of both stimuli (IC, *Caruso et al., 2018*), strictly fixation (MF, AL, V4; this study, *Caruso et al., 2018*; *Jun et al., 2022*), and attend-elsewhere tasks (MT, V1; this study, *Jun et al., 2022*). As mentioned above, fluctuating activity can be observed when two stimuli are within the same receptive field (MT) as well as when they are not (V1). A key limitation is that comparisons between datasets are likely to be affected by parameters such as the amount of data acquired. Future work in which conditions are held constant as much as possible across datasets will be needed before strong claims can be made about matters such as whether

fluctuating activity is more vs. less prevalent across different behavioral tasks, brain areas, and even sensory systems. For example, the variation in behavioral tasks but consistency of results across datasets suggests that attentional task performance (or, conversely, lack of attentional control) is not a strong driver of the observed effects, but more work will be needed to fully establish this.

Our findings are part of an emerging body of literature suggesting that neural circuits fluctuate between different response states. For example, 'on/off' or 'up/down' states, characterized by different levels of spontaneous neural activity and/or responsiveness to stimuli, have been identified in multiple animal models, cortical areas, and behavioral contexts (*Hasenstaub et al., 2007*; *Harris and Thiele, 2011*; *Engel et al., 2016*). What causes these fluctuating activity patterns is not fully known, but they have been found to be influenced by attention and state of arousal (*Harris and Thiele, 2011*; *Engel et al., 2016*). The relationship between such up/down states and the fluctuations we have observed here remains unexplored. A key question is how the fluctuating activity evoked by multiple stimuli is coordinated across neurons. Up/down states can be either synchronized or desynchronized across neurons, depending on the behavioral context (*Harris and Thiele, 2011*; *Engel et al., 2016*). In our previous work, we showed that the fluctuating activity in V1 showed both positive and negative correlations across neurons in a fashion that suggests that at least some portion of the neural population responds to each stimulus on any given presentation (*Jun et al., 2022*). The neurons in the adjacent gratings MT dataset in our current study were recorded at the same time as the V1 dataset in that earlier study, so we can infer that the MT neurons would also likely have shown both positive and negative correlations with those simultaneously recorded V1 neurons.

Also relevant are studies of rhythmicity and/or serial sampling in attention (e.g., *Treisman and Gelade, 1980*; *Van Rullen, 2016*; *Fiebelkorn and Kastner, 2019*). Reaction times to detect a target among distractors can scale with the number of distractor stimuli (depending on how the target differs from the distractors) (*Treisman and Gelade, 1980*). Near threshold stimuli or stimulus changes occurring at certain phases of electroencephalography (EEG) or local field potential (LFP) oscillatory cycles are more likely to be detected than those occurring at other phases (*Busch et al., 2009*; *Busch and VanRullen, 2010*; *Vanrullen et al., 2011*; *Fiebelkorn et al., 2013*; *McLelland and VanRullen, 2016*; *Fiebelkorn et al., 2018*; *Helfrich et al., 2018*). And conversely, rhythmic fluctuations can be triggered by presentation of multiple stimuli in V4 (*Kienitz et al., 2018*; *Kienitz et al., 2022*).

Collectively, these findings support the view that aspects of the nervous system sequentially samples the sensory environment across time, a notion with a clear conceptual link to the findings presented here. Future work will be needed to fully explore this possible connection, for several reasons. First, our current analysis method does not actually require fluctuating activity to occur rhythmically so long as it fluctuates. Thus, our method may be uncovering patterns of fluctuating activity that have not been seen before, due to irregularity in the timing of fluctuations or their coordination across the neural population. Second, our method requires that the fluctuating activity be reasonably well modeled as fluctuating between two specific benchmarks, namely the activity evoked by stimulus A vs. B. A combined analysis method that evaluates the periodicity of such benchmarked fluctuating activity will be useful for bridging the gap between the present study and these related findings. In addition, deployment of attentional tasks will be needed to fully explore this connection.

Another area for future exploration is the extent to which fluctuating responses could allow for preservation of more than two stimuli. The current study exclusively used datasets in which two stimuli were presented, either alone or in pairs. It will be sensible to extend these analysis methods to more complex but realistic stimulus conditions involving three or more stimuli.

In sum, neural switching appears to be a general phenomenon throughout a range of areas in the visual system and could support the ability to preserve information about multiple sensory components. Fluctuating activity may relate to object formation, a key perceptual ability whose underlying mechanism(s) remain poorly understood. Future experimentation is needed to understand how neural fluctuations may support such cognitive phenomena.

## Acknowledgements

We thank Justine Shih, Tingan Zhu, Gelana Tostaeva, Justine Griego, and Sara Gannon for helpful discussions about these topics. This work was supported by F31 fellowship to MNS; National Institutes of Health grant nos. R00EY020844 (MRC), R01EY022930 (MRC); Core Grant P30 EY008098s

(MRC); R01DC013906 (JMG, STT); and R01DC016363 (JMG, STT); and by support to MRC from the McKnight, Whitehall, Sloan, and Simons Foundations.

## Additional information

### Competing interests

Jennifer M Groh: Reviewing editor, *eLife*. The other authors declare that no competing interests exist.

### Funding

| Funder | Grant reference number | Author |
|---|---|---|
| National Institute on Deafness and Other Communication Disorders | F31DC020361 | Meredith N Schmehl |
| National Eye Institute | R00EY020844 | Marlene Cohen |
| National Eye Institute | R01EY022930 | Marlene Cohen |
| National Eye Institute | P30EY008098 | Marlene Cohen |
| National Institute on Deafness and Other Communication Disorders | R01DC013906 | Surya T Tokdar Jennifer M Groh |
| National Institute on Deafness and Other Communication Disorders | R01DC016363 | Surya T Tokdar Jennifer M Groh |

The funders had no role in study design, data collection, and interpretation, or the decision to submit the work for publication.

### Author contributions

Meredith N Schmehl, Conceptualization, Data curation, Formal analysis, Funding acquisition, Validation, Investigation, Visualization, Methodology, Writing – original draft, Project administration, Writing – review and editing; Valeria C Caruso, Conceptualization, Data curation, Formal analysis, Validation, Investigation, Visualization, Methodology, Writing – original draft, Project administration, Writing – review and editing; Yunran Chen, Conceptualization, Data curation, Software, Investigation, Methodology, Writing – review and editing; Na Young Jun, Data curation, Formal analysis, Investigation, Methodology, Writing – review and editing; Shawn M Willett, Jeff T Mohl, Douglas A Ruff, Akinori F Ebihara, Conceptualization, Data curation, Investigation, Methodology, Writing – review and editing; Marlene Cohen, Winrich A Freiwald, Conceptualization, Data curation, Supervision, Funding acquisition, Investigation, Methodology, Writing – review and editing; Surya T Tokdar, Conceptualization, Data curation, Software, Formal analysis, Supervision, Funding acquisition, Validation, Investigation, Methodology, Writing – review and editing; Jennifer M Groh, Conceptualization, Data curation, Formal analysis, Supervision, Funding acquisition, Validation, Investigation, Visualization, Methodology, Writing – original draft, Project administration, Writing – review and editing

### Author ORCIDs

Meredith N Schmehl  http://orcid.org/0000-0002-0848-309X
Valeria C Caruso  http://orcid.org/0000-0002-3895-1296
Na Young Jun  http://orcid.org/0000-0002-8841-3947
Marlene Cohen  http://orcid.org/0000-0001-8583-4300
Winrich A Freiwald  http://orcid.org/0000-0001-8456-5030
Jennifer M Groh  http://orcid.org/0000-0002-6435-3935

### Ethics

All datasets were collected in awake, behaving macaques (MT: Macaca mulatta; IT: Macaca mulatta and Macaca fascicularis). The data were originally collected at the University of Pittsburgh (Ruff and Cohen, 2016; Ruff et al., 2016), Rockefeller University (Ebihara, 2015), and the German Primate Center in Goettingen (Li et al., 2016). All procedures were conducted in accordance with applicable US,

German, and European guidelines or regulations and were approved by the relevant compliance committees at the relevant institutions (US guidelines: National Institutes of Health (NIH) Pub. No. 86-23, Revised 1985; Pittsburgh MT grating datasets: Institutional Animal Care and Use Committees of the University of Pittsburgh and Carnegie Mellon University (Protocol #: 20067560, PHS Assurance Number: D16-00118); IT datasets: The Rockefeller University Institutional Animal Care and Use Committee (Protocol #: 21104-H USDA, PHS Assurance Number: A3081-01); Goettingen MT dot patch dataset: [Niedersx00E4;chsisches Landesamtfx00FC;r Verbraucherschutzund Lebensmittelsicherheit (LAVES)] permit numbers 33.42502/08-07.02 and 33.14.42502-04-064/07). All surgical procedures were conducted aseptically and with suitable anesthetics and analgesics.

### Decision letter and Author response

Decision letter https://doi.org/10.7554/eLife.91129.sa1

Author response https://doi.org/10.7554/eLife.91129.sa2

## Additional files

### Supplementary files

• MDAR checklist

• Source data 1. Source data for the figures and analyses in this article. Spreadsheets are included for the analysis output from each dataset. Individual file names indicate which datasets they relate to.

### Data availability

Some of the data used in this study has been published previously (https://doi.org/10.5061/DRYAD.88PV1). Data underlying the figures can be found in Source data 1.

The following previously published dataset was used:

| Author(s) | Year | Dataset title | Dataset URL | Database and Identifier |
|---|---|---|---|---|
| Li K, Kozyrev V, Kyllingsbæk S, Treue S, Ditlevsen S, Bundesen C | 2020 | Data from: Neurons in primate visual cortex alternate between responses to multiple stimuli in their receptive field | https://doi.org/10.5061/dryad.88pv1 | Dryad Digital Repository, 10.5061/dryad.88pv1 |

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
