## [Editor Report]

This important study adds to the growing body of evidence that neural responses fluctuate in time to alternatively represent one among multiple concurrent stimuli and that these fluctuations seize when objects fuse into one perceived object. The present study provides solid evidence from multiple brain areas and stimuli types to support this hypothesis. Overall, the study illustrates how the brain can use time dimension and synchrony to either parse or integrate stimuli into a coherent representation.

---

## [Decision Letter]

**Decision letter after peer review:**

Thank you for submitting your article "Multiple objects evoke fluctuating responses in several regions of the visual pathway" for consideration by *eLife*. Your article has been reviewed by 2 peer reviewers, and the evaluation has been overseen by a Reviewing Editor and Joshua Gold as the Senior Editor.

Essential revisions:

Please revise to more appropriately place the work in the context of prior studies and address possible confounds from differences in attention and statistical analyses related to selection criteria for microsaccades and spike train smoothing.

*Reviewer #1 (Recommendations for the authors):*

Attention state: On p. 5. It is stated that the "stimuli in question … were unattended". Could you elaborate how this was controlled? When maintain central fixation under passive viewing conditions, stimuli presented in the periphery are typically not ignored, but at least partially attended (unless one engages the participant in a highly demanding task at fixation). How did attention state factor into the results given the work of Engel and Moore on fluctuating firing patterns due to attentional state?

Microsaccades: Given that microsaccades (MSs) might play an important role in the modulation of firing patterns it is important to control for them. MSs were defined as "time periods during which eye speed exceeded 6 standard deviations". Can you present a control analysis to show what this practically means? MSs have standard definitions and occur along a continuum. It will be important to understand which part of the continuum is excluded by your definition.

*Reviewer #2 (Recommendations for the authors):*

The size of receptive fields in MT and IT may vary quite a bit. The optimal stimulus size may therefore be important information for the reader. To give a better sense of the stimuli used in these experiments, please add scale bars to panels in Figure 1.

Please clarify the time window on the x-axis when showing spike counts, such as in Figure 4. Can this be interpreted as spikes/second, or converted into spikes/second when analyzing a shorter time window? This is especially relevant when looking at the overall spike counts in relation to other studies. Whereas some previous work has shown vigorous spiking for faces in IT face patches, the spike counts for preferred faces of ~5 spikes seems very low. However, if this is not yet converted into spikes/second, it may be difficult to compare. Note: there may be good reasons to stick to total spike count instead of spike rate. If so, please just clarify this in the figure caption to give the reader an idea of what the spike rate may be.

Unless this goes against the journal guidelines, please improve the readability of statistical results by rounding the p-values such that they are not a long list of zeros (Figure 3e) or overly precise (e.g. Figure 4g).

---

## [Author Response]

Essential revisions:Please revise to more appropriately place the work in the context of prior studies and address possible confounds from differences in attention and statistical analyses related to selection criteria for microsaccades and spike train smoothing.

We are grateful for all the reviewer comments and the “license” to expand the scope of the literature to which we think this work relates. We have expanded the introduction and discussion to incorporate additional coverage of exciting relevant literature in the domains of attention and fluctuating states in the nervous system as revealed by other studies. We have also clarified procedures related to spike train smoothing and incorporated raw data into the relevant figures. With regard to microsaccades, we have addressed omissions in the methods section regarding fixation control more generally, and we present below a more detailed summary of relevant parameters and previous analyses.

Reviewer #1 (Recommendations for the authors):Attention state: On p. 5. It is stated that the "stimuli in question … were unattended". Could you elaborate how this was controlled? When maintain central fixation under passive viewing conditions, stimuli presented in the periphery are typically not ignored, but at least partially attended (unless one engages the participant in a highly demanding task at fixation). How did attention state factor into the results given the work of Engel and Moore on fluctuating firing patterns due to attentional state?

We appreciate this comment – we have revised the text to indicate that the monkeys were directed to fixate away from the stimuli in question, but we recognize that eye position may not be informative about where attention was directed during each trial. We have rewritten this section to clarify:

“Our goal was to assess fluctuating neural activity in multiple visual areas and in response to multiple stimuli (Table 1). In all datasets, fixation was controlled while the stimuli were presented (0.5-1.2 degree fixation window radius, depending on the dataset). In some datasets, monkeys performed a behavioral task in which attention was directed away from the stimuli, such as to the fixation position itself or to an additional stimulus in the ipsilateral hemifield. In the fixation-only tasks, it should be noted that the absence of experimental control over attention could have led to the monkeys covertly attending the receptive field stimuli from time to time.” (Methods, Introduction to Datasets, para 2).

Nevertheless, we do not think that a varying locus of attention during fixation can fully account for fluctuating activity. If this were the case, we would have expected to see fluctuating activity in all the datasets that involved passive fixation, including the MT “superimposed gratings” dataset, but we do not. Accordingly, we now include this argument in the discussion, while acknowledging that the ideal scenario would be to deliberately test the findings in an attention task:

“For example, the variation in behavioral tasks but consistency of results across datasets suggests that attentional task performance (or, conversely, lack of attentional control) is not a strong driver of the observed effects, but more work will be needed to fully establish this.” (Discussion, paragraph 6)

Microsaccades: Given that microsaccades (MSs) might play an important role in the modulation of firing patterns it is important to control for them. MSs were defined as "time periods during which eye speed exceeded 6 standard deviations". Can you present a control analysis to show what this practically means? MSs have standard definitions and occur along a continuum. It will be important to understand which part of the continuum is excluded by your definition.

We address this comment at both the specific and conceptual levels:

First, at the conceptual level, given that our analysis focused on trial-by-trial spike counts, the main concern would be fluctuations in fixation across trials; any affects of microsaccades within trials would likely muddy this analysis but not serve as a confound (especially given the short duration of the spike counting windows – a mere 200 ms does not allow much time for a stimulus-triggered microsaccade to cause the stimulus to land on a different portion of the retina and be reflected in the spike count on that trial).

That said, this comment made us aware that we had failed to include information about the size of the fixation windows for the datasets included in this manuscript. We have now included these details in the methods section, first as an overall range:

“In all datasets, fixation was controlled while the stimuli were presented (0.5-1.2 degree radius, depending on the dataset)”, (Methods, Introduction to datasets para 2),

and then again where the specific datasets and tasks are described (see tracked changes). Any saccade that caused the eyes to depart from these windows during stimulus presentation would have caused the trials to be excluded.

Indeed, the actual variation in fixation was probably considerably smaller than these fixation windows. While we do not have the eye tracking data for all datasets, in Jun et al. *eLife* 2018 we did analyze the fixation scatter for the V1 recordings that occurred simultaneously with the MT adjacent gratings dataset in the present study.

In short, the average deviation of the eyes from the fixation target was less than 0.15 degrees – a tiny amount given the size of MT receptive fields.

With regard to screening microsaccades specifically, we agree that we should have included more information about how this was done and the precedent for it. The precedent for this screening comes from Engbert and Kliegl (2003), in which microsaccades were defined by eye speeds that exceed 6 standard deviations from the mean speed. We also deployed this method previously in our prior publication involving other recordings from these same sets of experiments (Jun et al. *eLife* 2022). We have expanded the description in “Methods/ Analysis: response periods and inclusion criteria/ Trial inclusion criteria” to explain this more fully:

“Microsaccades were defined as periods of time in which eye speed exceeded the mean eye speed in the fixation epoch by more than 6 standard deviations following standards established by (Engbert and Kliegl, 2003) and deployed in our previous studies involving these same experiments (Jun et al. 2022, Ruff et al. 2016, Ruff and Cohen 2016)”.

Reviewer #2 (Recommendations for the authors):The size of receptive fields in MT and IT may vary quite a bit. The optimal stimulus size may therefore be important information for the reader. To give a better sense of the stimuli used in these experiments, please add scale bars to panels in Figure 1.

We thank the reviewer for this suggestion to increase clarity, and we have added scale bars to this figure, with the exception of the MT dot patch dataset, for which the stimulus sizes were not provided in the original Li et al. study.

Please clarify the time window on the x-axis when showing spike counts, such as in Figure 4. Can this be interpreted as spikes/second, or converted into spikes/second when analyzing a shorter time window? This is especially relevant when looking at the overall spike counts in relation to other studies. Whereas some previous work has shown vigorous spiking for faces in IT face patches, the spike counts for preferred faces of ~5 spikes seems very low. However, if this is not yet converted into spikes/second, it may be difficult to compare. Note: there may be good reasons to stick to total spike count instead of spike rate. If so, please just clarify this in the figure caption to give the reader an idea of what the spike rate may be.

For all analyses, we counted spikes in a window of 200 ms, shifted by the approximate response latency of each brain region (30 to 230 ms after stimulus onset for MT, and 50 to 250 ms after stimulus onset for AL and MF). Therefore, the spike counts shown in these figures should be multiplied by a factor of 5 to give their equivalent values in spikes/second. We have clarified this in the captions of Figures 3 and 4, e.g.:

“Spike counts were computed for a window 200 ms in duration (see Methods); multiply x-axis by 5 to convert to spike rates in Hz.”

Unless this goes against the journal guidelines, please improve the readability of statistical results by rounding the p-values such that they are not a long list of zeros (Figure 3e) or overly precise (e.g. Figure 4g).

Thank you for catching this. We have rounded the p-values in Figure 4G and Figure 4—figure supplement 1F to two significant digits, which now matches the text. We have also made adjustments to the graphics of these figures so that it is more clear which comparisons these p values relate to. For now, we are leaving the “long list of zeros” p values, which involved the MT adjacent vs. superimposed gratings chi-square test. Technically, the p value returned by the chi-square function in Matlab was zero, with a floating point precision suggesting a value of less than 2.22507e-308. However, we are skeptical that the chi-square test can provide that level of significance for a dataset of this size; we will seek guidance from the journal with regard to how to convey this p value according to their standards.